# High-Resolution Multisensor Remote Sensing to Support Date Palm Farm Management

**Maggie Mulley** [1,*,†], **Lammert Kooistra** [1] **and Laurens Bierens** [2] 

[1]  Laboratory of Geo-Information Science and Remote Sensing, Wageningen University and Research, P.O. Box 47, 6700 AA Wageningen, The Netherlands; lammert.kooistra@wur.nl
[2]  TEC-IB B.V., Oude Veiling 29, 2635 GK Den Hoorn (ZH), The Netherlands; l.bierens@tec-ib.com
*  Correspondence: maggie.m.mulley@gmail.com
†  Current employment: Aerovision BV, Stadsring 47, 3811 HN Amersfoort, The Netherlands.

**Abstract:** Date palms are a valuable crop in areas with limited water availability such as the Middle East and sub-Saharan Africa, due to their hardiness in tough conditions. Increasing soil salinity and the spread of pests including the red palm weevil (RPW) are two examples of growing threats to date palm plantations. Separate studies have shown that thermal, multispectral, and hyperspectral remote sensing imagery can provide insight into the health of date palm plantations, but the added value of combining these datasets has not been investigated. The current study used available thermal, hyperspectral, Light Detection and Ranging (LiDAR) and visual Red-Green-Blue (RGB) images to investigate the possibilities of assessing date palm health at two "levels"; block level and individual tree level. Test blocks were defined into assumed healthy and unhealthy classes, and thermal and height data were extracted and compared. Due to distortions in the hyperspectral imagery, this data was only used for individual tree analysis; methods for identifying individual tree points using Normalized Difference Vegetation Index (NDVI) maps proved accurate. A total of 100 random test trees in one block were selected, and comparisons between hyperspectral, thermal and height data were made. For the vegetation index red-edge position (REP), the R-squared value in correlation with temperature was 0.313 and with height was 0.253. The vegetation index—the Vogelmann Red Edge Index (VOGI)—also has a relatively strong correlation value with both temperature ($R^2 = 0.227$) and height ($R^2 = 0.213$). Despite limited field data, the results of this study suggest that remote sensing data has added value in analyzing date palm plantations and could provide insight for precision agriculture techniques.

**Keywords:** remote sensing; date palms; precision agriculture; plantation management; thermal; hyperspectral

---

## 1. Introduction

Across the globe, people are suffering from a lack of food availability and food security as populations grow and land becomes unsuitable for farming [1–3]. There is a need for crops with high nutritional value that can withstand the arid and semiarid conditions in countries with limited water resources. The date palm is one such plant [2]. In the Middle East and North Africa alone, 100 million palms are cultivated on one million hectares of cropland [4].

There are several biotic and abiotic factors that affect the management of a healthy date palm crop. Global climate change may pose a threat to plantations of date palm; climate models predict that regions suitable for date palm growth will shrink, especially in the Middle East [5]. Water is a limiting factor for growth, although excess water can also reduce yield. Studies investigating the effects of water on date palm growth have found that insufficient water application slows the growth of the plants [6,7].



Increasing salinity also appears to negatively impact the growth of date palms [8–10]. This is a growing problem in arid regions as groundwater reservoirs are depleted due to excess irrigation.

Plantations of date palm and other palm species are also under threat due to infestations by different pests, the most extensive of which is the red palm weevil, or *Rhynchophorus ferrugineus* (Olivier, 1790). *R. ferrugineus* is spread via the transport of cuttings of palm tree leaves [4,11], and has been detected in 50% of date palm growing countries [12]. The effects of *R. ferrugineus* are severe; in the later stages of infestation the trunk can be damaged to the extent that the tree collapses and will most certainly die. Other pests that affect date palms include the Dubas bug, lesser date palm moth, and many different borer species [13]. The Dubas bug feeds on leaf sap which results in yellowing and wilting of leaves [3]. Lesser date palm moths infest and damage the fruit of the date palm [13]. Borers have similar effects to *R. ferrugineus*, although they may infest different parts of the tree, such as the middle rib of the frond and the upper section of the tree near the bunches of fruit [13].

At present, a heavy focus is placed on monitoring plantations for *R. ferrugineus* infestation. Unfortunately, current methods of detection do not allow for effective, reliable identification of infested trees [14]. In fact, the detection of palms in the early stages of infestation has been stated as the main issue in controlling weevil infestation [15]. It is very difficult to recognise whether a tree is infested based on visual inspection as the symptoms vary depending on conditions such as the tree species and initial site of infestation [14,16]. All of the current detection methods, including acoustic detection, x-ray, and sniffer dogs are often ineffective on a large scale because each tree has to be inspected individually [14,16].

Remote sensing has been suggested as a potential detection method, as it allows for large scale monitoring and effective data analysis techniques [16]. Thermal imaging studies have shown it might be possible to detect plants infested with *R. ferrugineus* using aerial thermal photographs [17]. Multi- and hyperspectral images have successfully identified infested trees. WorldView-3 satellite imagery has been used for *R. ferrugineus* detection [18]. Hyperspectral imagery was conducted on oil palms to classify *R. ferrugineus*-infested palms based on vegetation indices [19]. Light Detection and Ranging (LiDAR) has not been applied to the palm assessment case, but it does have potential; for example, LiDAR has been used to classify eucalyptus tree health [20].

Remote sensing has also been applied to assess the effect of other stressors on palm plantations. Cohen et al. (2012) used thermal images to investigate how changes in irrigation affect the health of date palm plantations [6]. The effect of soil salinity on date palm health has been assessed by using Landsat Thematic Mapper (TM) and Enhanced Thematic Mapper Plus (ETM+) images to calculate the Soil Adjusted Vegetation Index (SAVI) for two different years [21].

The studies listed above have all explored how individual stressors affect date palm plantations. However, it is highly likely that a date palm plantation will be subject to multiple stressors at once and identifying the influence of these effects will allow for more comprehensive and efficient plantation management. Remote sensing approaches can provide the tools to conduct this form of management; analysis of the entire plantation is possible using automated techniques in order to differentiate between a combination of stressors.

Therefore, the goal of this study is to explore which remote sensing sensors and derived indicators can detect stressors on palm health at different spatial levels, in order to support a holistic solution for effective date palm management. Plantations are often organised in large rectangular sub-sections or blocks—this will be the first spatial level to be evaluated. The second level of analysis will be conducted on an individual tree level. For this study, several different remote sensing imagery sources have been used, in order to determine whether different techniques are most suitable for highlighting differences in palm status. The available imagery sources are thermal, hyperspectral, and LiDAR, with data collected using airborne platforms. High resolution imagery is achievable with airborne platforms, allowing for analysis of individual tree health.

## 2. Materials and Methods

### 2.1. Study Area and Data Description

The farm shown in Figure 1 is in the Al-Kharj region in Saudi Arabia, southeast of the capital Riyadh. Saudi Arabia was one of the earliest countries growing date palms to be affected by *R. ferrugineus* infestation outside of South East Asia, with first infestations occurring since the 1980s [22]. The country has the highest per capita consumption of dates in the world. Flood irrigation is making way for more modern irrigation methods such as drip irrigation in many farms [23].

Of the farm, an area of 168.8 hectares (outlined in blue in Figure 1) has been chosen as the key study area of this analysis. This farm is relatively well managed, with continuous maintenance and targeted irrigation practices in place. As is exemplified in Figure 1b, date palm plantations are commonly divided into regular-sized rectangles of approximately 10 hectares; these sub-sections are hereafter referred to as blocks. Figure 1c illustrates the uniform growth patterns of the date palms; the crowns are distinct and organised in a grid-like structure.

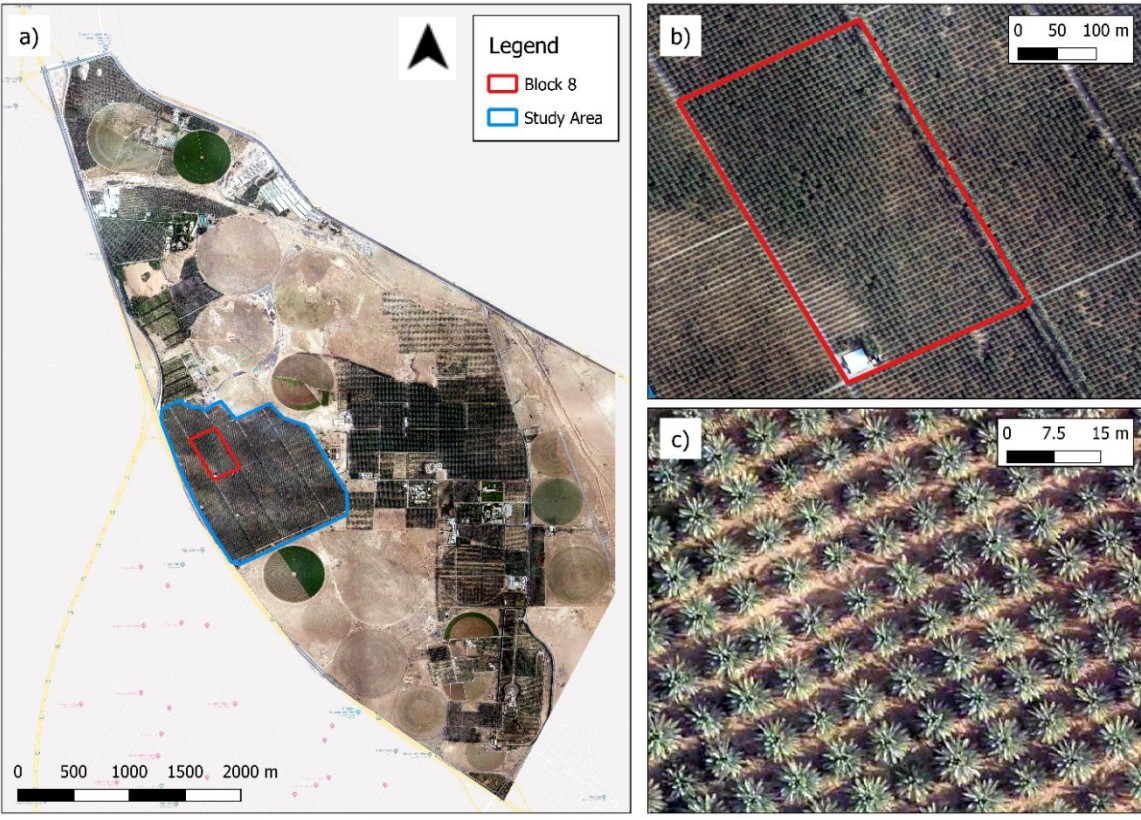

**Figure 1.** Visual red-green-blue (RGB) images with 15-cm resolution, taken using an aerial platform on 26 May 2016, of (**a**) the date palm plantation in the study area (outlined in blue), located south-east of Riyadh, the capital of Saudi Arabia; (**b**) an example of a block within the plantation (outlined in red) and (**c**) an example of individual palm trees.

For this study, four remote sensing data types were available for analysis: hyperspectral, thermal, LiDAR, and high-resolution visual red-green-blue (RGB) images. Each dataset contains different information that will be useful when assessing date palm health. The data has been collected using a Diamond DA42 MPP GEOSTAR (Diamond Aircraft Industries GmbH, Wiener Neustadt, Austria) aircraft, which is designed to carry many different remote sensing instruments. The flight was conducted on 26 May 2016 between approximately 6:00 and 9:00AM.

Table 1 shows information about the four sensors used for data collection. The spatial resolution refers to the pixel size of the data used in the analysis after preprocessing; the original LiDAR data was provided as a point cloud but has been processed into raster data.

**Table 1.** Characteristics of the four sensor systems used for data collection.

| Data Type | RGB | LiDAR | Thermal | Hyperspectral | |
| --- | --- | --- | --- | --- | --- |
| Name of sensor [1] | Phase One iXA-R Camera [2] | RIEGL LMS-Q1560 [3] | VarioCAM®HD head [4] | HySpex VNIR-1800 [5] | Hy-Spex SWIR-384 [5] |
| Type of sensor | Medium-format camera system | Rotating polygon mirror | Uncooled microbolometer focal-plane array | Pushbroom camera actively cooled and stabilized scientific CMOS detector | Pushbroom camera Mercury cadmium telluride sensor |
| Spectral range | Visible | Near-infrared | 7.5–14 µm | 0.4–1 nm, 182 bands | 0.93–2.5 nm, 288 bands |
| Spatial resolution | 0.15 m, 0.6 m, 1.8 m | 1 m, 2 m | 1.8 m | 1 m | 1 m |

[1] Manufacturer information; [2] Phase One Industrial, Frederiksberg, Demark; [3] RIEGL Laser Measurement Systems GmbH, Horn, Austria; [4] InfraTec, Dresden, Germany; [5] Norsk Elektro Optikk AS (NEO), Skedsmokorset, Norway.

*2.2. Methodology*

2.2.1. Preprocessing

Different preprocessing steps were necessary for each of the multiple remote sensing data sources available. These steps will be described briefly in the next few paragraphs. Steps were carried out in various software: R (version 3.3.1, R Foundation for Statistical Computing, Vienna, Austria), QGIS (version 2.1.4 'Essen', Open Source Geospatial Foundation, Chicago, IL, USA), ArcGIS (version 10.4, ESRI, Redlands, CA, USA), LAS tools (version 2016, rapidlasso GmbH, Gilching, Germany), and Agisoft PhotoScan (version 1.2, Agisoft LLC, St. Petersburg, Russia).

The RGB data was provided in JPEG format and was not geographically referenced. The files were given a geographic coordinate system and then projected to WGS 1984 UTM Zone 38N, and also converted to GeoTIFF. This projection was also used for all the other datasets.

The LiDAR dataset was originally received as a point cloud. To generate a tree height raster, Digital Surface Models (DSM) at different resolutions was generated; additionally, the mean, minimum, maximum and standard deviation values of the points within these pixels were used.

During acquisition, the VarioCAM thermal camera was controlled using the IRBIS®Thermography Software (InfraTec, Dresden, Germany). Within this software, the temperature range for acquisition was set to values between 0 and 50 degrees Celsius [24]. This range was broader than the actual temperature range which could be expected but was set by the flight operators to avoid saturation at low or high values. The raw thermal data was provided as individual snapshot images that were then stitched together using the structure-from-motion method [25] as implemented through Agisoft PhotoScan. In some areas where there are a lot of overlapping images, tie-points do not match perfectly, causing some blurring in the image; where this occurs, individual trees are not identifiable. This effect cannot be reduced without compromising the resolution of the entire image. The image was received as a (8 bit) greyscale image with 256 levels. To convert the values into a relative approximation of degrees Celsius, the acquired greyscale image was converted using the multiplication factor of 0.1953125 to fit the temperature range settings during acquisition (0 and 50 degrees Celsius).

The hyperspectral images were received without the inertial measurement unit (IMU) data due to an error in the acquisition procedure. This means that it was not possible to correct for image distortion due to plane movement. The images were also provided in an unknown multiplication factor, and thus the values are unusual. However, an evaluation of this data source is still considered worthwhile

as it was acquired at the same time as the other remote sensing types. This meant that extensive steps were required to attain usable imagery.

By using the recorded times and positioning of the RGB imagery, the order in which hyperspectral images were acquired was determined. This meant that a representative image could be constructed and aligned with the study area. The image was then transformed using digitally assigned geometric control points as identified from the RGB images. The QGIS *Georeferencer* tool was used, with transformation type set as Thin Plate Spline which is useful for heavily distorted data, and Nearest Neighbour as a resampling method to maintain the original data values. The pixel data was multiplied by a scaling factor of 0.002 as this ensured they were in the relative range of standard reflectance values. As this is not the standard way to determine actual reflectance values, it is only possible to carry out relative comparisons within the dataset—this dataset cannot be numerically compared with any hyperspectral data collected in later studies.

### 2.2.2. Identification of Blocks and Trees

As analysis is carried out on two different spatial levels, namely blocks and trees, it is necessary to identify these regions. Block shapefiles were provided for the farm, and these were used for analysis of the thermal and LiDAR-derived height data. As the hyperspectral blocks were geometrically distorted, these were drawn manually.

Several methods were used in order to identify the trees for tree-level analysis. For the non-distorted data, the canopy height map as determined from LiDAR was used to locate individual tree points. This was done using the watershed method, which involves inverting the height rasters, calculating the sinks and converting them to points, and filtering these points at an appropriate buffer width. The software used was LAS tools and ArcMap.

It was also decided that deriving canopy polygons could be interesting as the area of the tree canopy could prove a useful indicator of tree health. The LiDAR data was analysed using an R package called Individual Tree Crown (ITC) Segment (version 0.6, Michele Dalponte, San Michele all'Adige, Italy) [26]. This tool selects local maxima within a height raster and then spreads outwards to select the surrounding pixels that are within a specific height range representing the canopy. A polygon shapefile is generated, for which the height and area are calculated. A 3-m width limit was selected to constrain the growth of the polygon outside of the general tree size. To investigate sensitivity to spatial resolution, the model was run for pixel sizes of both 1 m and 2 m. In this case, the raster's containing the maximum value of the point cloud within each pixel were used for analysis, as these data were available for both 1-m and 2-m resolution.

For the hyperspectral data, a different method for tree identification was used. This is necessary because the distortions in the hyperspectral dataset mean that it is not possible to use the same tree locations for each dataset. It was decided to use vegetation indices, specifically normalised difference vegetation index (NDVI) (see Equation (1)), to identify the trees. Once the NDVI was calculated, based on the values of Equation [1], the *TreeTopFinderTool* in the R package "Forest Tools" (version 0.1.5, Andrew Plowright, BC, Canada) was used to identify local maxima. It is hypothesised that the points where the NDVI is highest represent the top of the tree canopy. A width threshold of four was set, based on the upper limit of the average canopy size, which is 3 to 4 m. Additionally, a minimum NDVI value of 0.3 was set to ensure that maxima in treeless regions were excluded—this value was based on the bimodal distribution of the NDVI values that distinguishes between soil and tree points.

$$NDVI = \frac{NIR_{750} - R_{660}}{NIR_{750} + R_{660}} \qquad (1)$$

### 2.2.3. Block-Level Analysis

The six blocks shown in Figure 2 have been selected for further analysis. These blocks were chosen based on the attributes provided in Table 2.

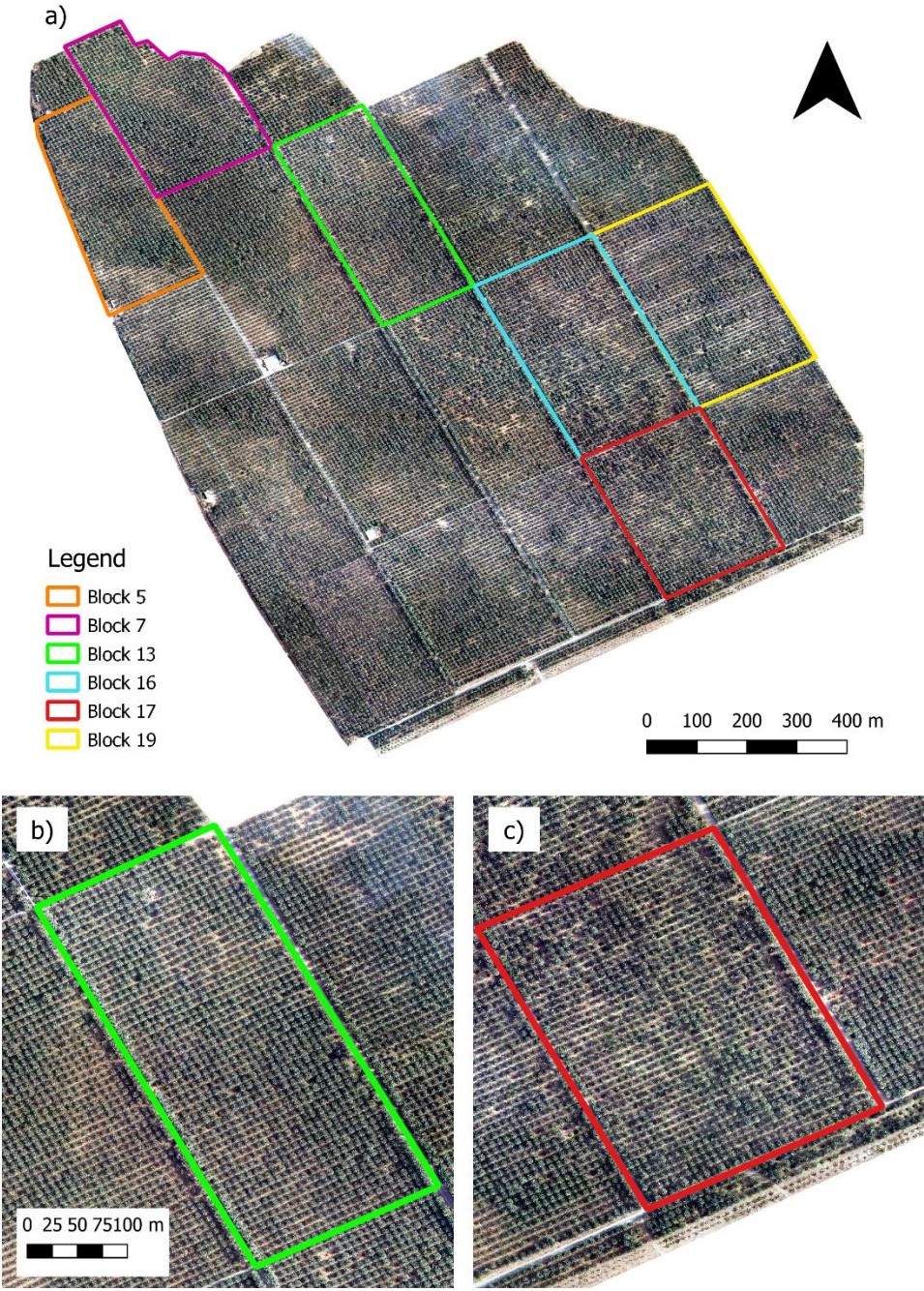

**Figure 2.** (**a**) Shows the selected test blocks; Blocks 5, 7, and 13 have been classified as healthy and Blocks 16, 17 and 19 have been classified as unhealthy. (**b**) and (**c**) shows an example of blocks classified as healthy (green) and unhealthy (red) respectively.

Two different groups of blocks have been determined based on their characteristics as described in Table 2. The data that was available from the farm manager mainly related to instances of *R. ferrugineus*, so this was a factor that was considered in block division. A visual assessment of the farm was made based on the homogeneity of canopy area and the sparseness of the block. The first group of trees was chosen with the requirements of having no instances of *R. ferrugineus* and of having high canopy homogeneity and low sparseness. The second group of blocks had relatively high instances of *R. ferrugineus,* and the appearance of the blocks revealed many gaps in trees and a wide variation of tree canopy widths. These two indicators were used to differentiate between potentially healthy and unhealthy groups. An additional factor that was considered was the amount of blurring in the thermal

image, as discussed in Section 2.2.1; if the blurring in a block was visually estimated as being more than 70%, it was not considered.

The assumed healthy test blocks are 5, 7, and 13, and the unhealthy blocks are 16, 17, and 19. Their respective values are shown in Table 2.

**Table 2.** Criteria used to determine the selection for the six test blocks.

| | **May 2016** | **June 2016** | **RGB Check** [1] | **Thermal Image Distortion** [2] |
|---|---|---|---|---|
| Block No. | No. RPW infested | No. RPW infested | 1–5 rating | Percent distorted |
| 5 | 0 | 0 | 1 | 20 |
| 7 | 0 | 0 | 1 | 30 |
| 13 | 0 | 0 | 2 | 30 |
| 16 | 8 | 4 | 5 | 30 |
| 17 | 10 | 4 | 5 | 30 |
| 19 | 1 | 9 | 5 | 20 |

[1] Indicator of block health status based on visual check for sparseness and heterogeneity, where 1 = homogenous and wide canopies, 5 = many patches and smaller canopies; [2] Visual check to determine how much of the image is affected by blurring due to structure-from-motion processing technique.

To summarise the statistical attributes of these blocks, boxplots were generated for temperature and height of the trees. These illustrate the spread of temperature and height across the test blocks, also in relation to all blocks in the farm. These boxplots illustrate the 25th, 50th and 75th percentiles, extreme values, and outlier values for the canopy pixels. For the temperature values, the derived canopy areas were used as a mask layer for the temperature raster. The height values are also based on the polygon shape. A visual check was conducted to explain the findings of the boxplot analysis.

For each block, a total of 10 'healthy' and 10 'unhealthy' trees were randomly selected after filtering based on their ratio between height and canopy area—based on discussions with the farm manager, it was hypothesised that trees with a high height area ratio (HAR), meaning they have a small canopy in relation to height, would be more likely to be unhealthy. Trees were selected based on whether they were above or below a threshold for this ratio and were also filtered based on height to attempt to select trees of similar height.

The mean temperatures of each group of trees (healthy and unhealthy) have been compared using two-factor analysis-of-variance (ANOVA) analysis. This statistical test is used to determine whether the differences in means of two or more groups at two or more levels are statistically significant; in this case, between the 'healthy' and 'unhealthy' trees, between the blocks, and the interaction effect between them. The post-hoc Tukey honest significant difference (HSD) test is applied following ANOVA—it is a comparison test that determines which pairs of means are responsible for the statistically significant test result. A significance level of $\alpha = 0.05$ has been used to determine whether the result is statistically significant or not.

### 2.2.4. Tree-Level Analysis

Different blocks were used for the individual tree analysis, as several of the blocks were not available in the hyperspectral dataset. Block 11 was selected for analysis for two reasons; firstly, it has the least visible blurring in the thermal dataset, and also it has a wide variety of trees with different canopy sizes.

As described in Section 2.2.2, the tree canopies have been identified based on NDVI. The next step was to select trees that can be used to extract more extensive information. In this case, trees were selected randomly from the hyperspectral dataset, and were matched manually to the trees in the thermal dataset. This was done by starting at the edges of the trees in each block and then counting vertically and the horizontally to identify the correct tree. A shapefile was created, and points were placed on the identified trees. A total of 100 trees was used for this block. As the trees are quite

noticeable in each dataset, it is expected that this method was quite accurate at detection, although in some cases the tree may be misidentified.

Fourteen vegetation indices (VI) have been selected for further analysis (See Appendix A Table A1). These VIs can be split into general groups that focus on different regions of in the visible light spectrum, such as green, red, and the red-edge position. For each test tree, the correlation between each vegetation index and temperature and height has been derived, and R squared values have been calculated. A visual check has also been carried out.

## 3. Results

### 3.1. Block and Tree Identification Results

The results of the method which identifies the areas of entire tree canopies can be seen in Figure 3. The ITC segment tool was run on two different height rasters (spatial resolutions of 1 m and 2 m) derived from the LiDAR dataset. Table 3 shows the number of tree points identified using the two different height raster maps; these values have been compared with tree numbers provided by the farm manager. For the 1 m resolution data these values exceeded 90% in every block. Fewer trees are detected using the 2 m resolution data: 73–90% were detected when comparing the number of tree polygons to the number of tree points in each block.

**Table 3.** Number (#) of trees within the test blocks based on the farm managers records compared to results of the Individual Tree Crown (ITC) segment analysis for 1 m and 2 m height rasters.

| Block No. | Recorded # of Trees | # of Trees (1 m Resolution) | # of Trees (2 m Resolution) |
|:---:|:---:|:---:|:---:|
| 5 | 1188 | 1137 | 1043 |
| 7 | 1213 | 1197 | 1084 |
| 13 | 1417 | 1277 | 1131 |
| 16 | 1749 | 1572 | 1242 |
| 17 | 1422 | 1302 | 1013 |
| 19 | 1691 | 1617 | 1440 |

When comparing Figure 3a,b, it can be seen that in the denser parts of the plantation, using the 1m resolution LiDAR data has a higher detection rate of tree canopies.

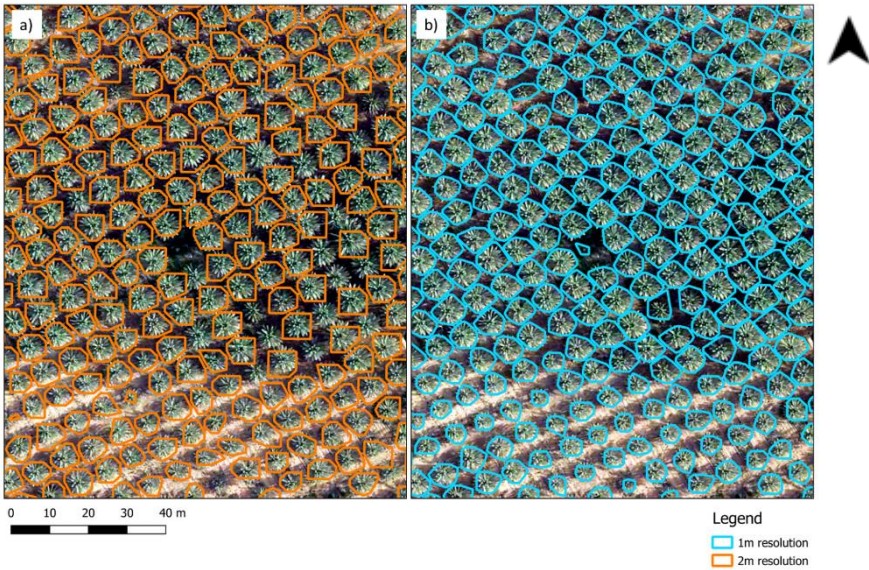

**Figure 3.** Section of an area of Block 5 in the farm. Canopy polygons derived using the ITC segment tool plotted in relation to the RGB data, for (**a**) the 2-m spatial resolution height raster (orange), and (**b**) 1-m spatial resolution (blue).

The second method used the NDVI image that was extracted from the hyperspectral dataset. Two different red bands were used to calculate the NDVI, and the comparisons can be seen in Figure 4. The outcome for both NDVI calculations is similar, although trees are identified more accurately in denser canopies when band 750 is used; for Block 16, for example, the number of trees recorded by the farm manager was 1749, compared to 1605 and 1601 for NDVI 750 and NDVI 800 respectively. For Block 19, the method using NDVI 750 identified 1661 trees, compared to 1656 (NDVI 800), and 1691 (trees recorded by farm manager). This indicates that in this case NDVI 750 is slightly more effective at identifying trees. Misidentification tends to be on the edges where the tree canopies are denser and therefore less distinct.

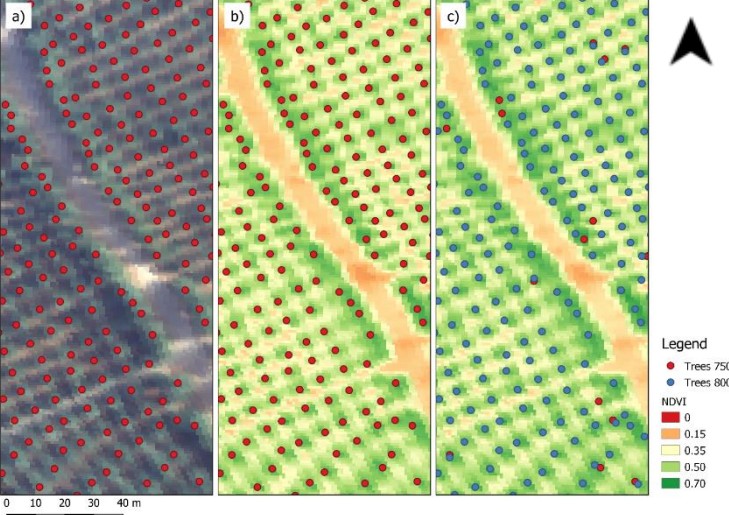

**Figure 4.** Tree points (red) derived using NDVI with band 750, overlaying (**a**) the hyperspectral image, and (**b**) the NDVI-derived using band 750, and (**c**) tree points derived using band 800 (blue) in comparison NDVI 750 points, overlying NDVI-derived using band 800.

For further analysis, the datasets with the highest detection rates were used. For analysis of the thermal and LiDAR datasets, the canopy polygons derived from the 1 m height rasters were used. The tree points resulting from the NDVI using spectral band 750 were chosen for the hyperspectral data analysis.

*3.2. Block-Level Analysis*

For block-level analysis, thermal data and height rasters were analysed. Six test blocks were chosen based on their attributes which have been discussed in the methods section. As the same six test blocks are not available for the hyperspectral dataset, this dataset has not been included in block-level analysis.

Boxplots of the thermal and height values for the six selected test blocks are presented in Figures 5 and 6. This illustrates the spread of heights across the test blocks, also in relation to all blocks in the farm. These boxplots illustrate the 25th, 50th and 75th percentiles, extreme values, and outlier values for the canopy pixels.

Figure 5 shows boxplots of the canopy temperature pixels of the test blocks. When treating block 5 as an outlier, it appears that two of the assumed healthy blocks have a mean temperature that is 0.5 °C lower than the unhealthy blocks. This is a positive result as the hypothesis was that unhealthy blocks would appear warmer than the healthy blocks. However, the mean temperature of block 5 is 0.5 °C higher than the mean temperature of the unhealthy test blocks. This could be because block 5 has been inaccurately classified as healthy. Feedback from the farm manager has indicated that the soil quality in this region is very poor so the trees there do not grow as well.

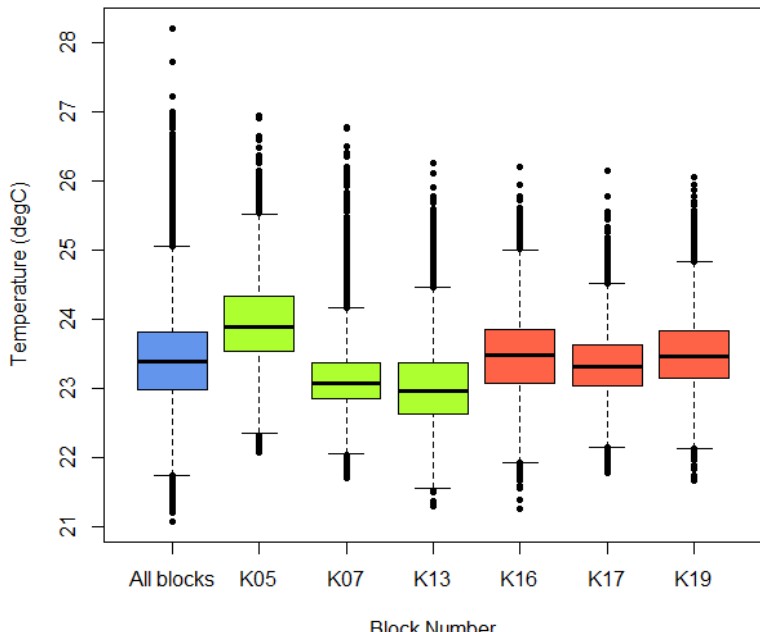

**Figure 5.** Boxplots of temperature for pixels defined as canopy using a buffer of 3 m and a height threshold of 4 m. The boxplot that is coloured blue shows the temperature for all blocks. The three test blocks in green (5, 7 and 13) are assumed healthy, and the three boxplots coloured red (16, 17 and 19) are unhealthy.

Figure 6 shows the outcome of the height analysis. The blocks classified as healthy appear to have similar heights to all blocks. Interestingly, the unhealthy blocks appear to deviate from the mean heights across all blocks. This is especially true for Block 19, which contains shorter trees than the other blocks. Block 16 appears to contain taller trees than the other blocks, and Block 17 has a wider spread within the upper and lower quartiles.

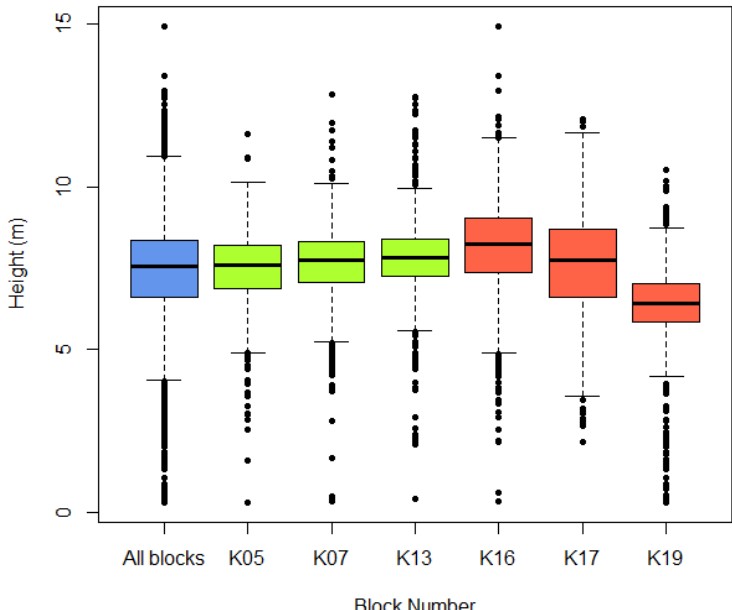

**Figure 6.** Boxplots for height of the blocks. The blue block shows the values for all blocks in the model farm. The healthy blocks are shown in green and the unhealthy blocks are shown in red.

The results of the boxplot analysis show some interesting trends, that are even more revealing when compared with the spatial plots of the blocks.

When looking at the spatial variation in the canopy pixels of the blocks (Figure 7), we can identify the areas that contribute to the varying values seen in the boxplots (Figure 5). Block 5 is evidently hotter than the other areas, especially in the south of the block, but also across the entire block as well. This is interesting, as the sparseness of the trees in the south of the block is similar to that in the sparse areas of the other blocks. It is difficult to determine the reason for this temperature difference. This area also corresponds to the high height area ratio (HAR) values seen in Figure 8. The unhealthy blocks clearly have more areas that are hotter. When comparing to the RGB data, these hotter areas tend to lie where the trees are sparser and thus more bare ground is exposed. There are also areas in the healthy blocks that have higher temperatures—this is because there is a high variability in all blocks and even those classified as healthy have unhealthy looking regions.

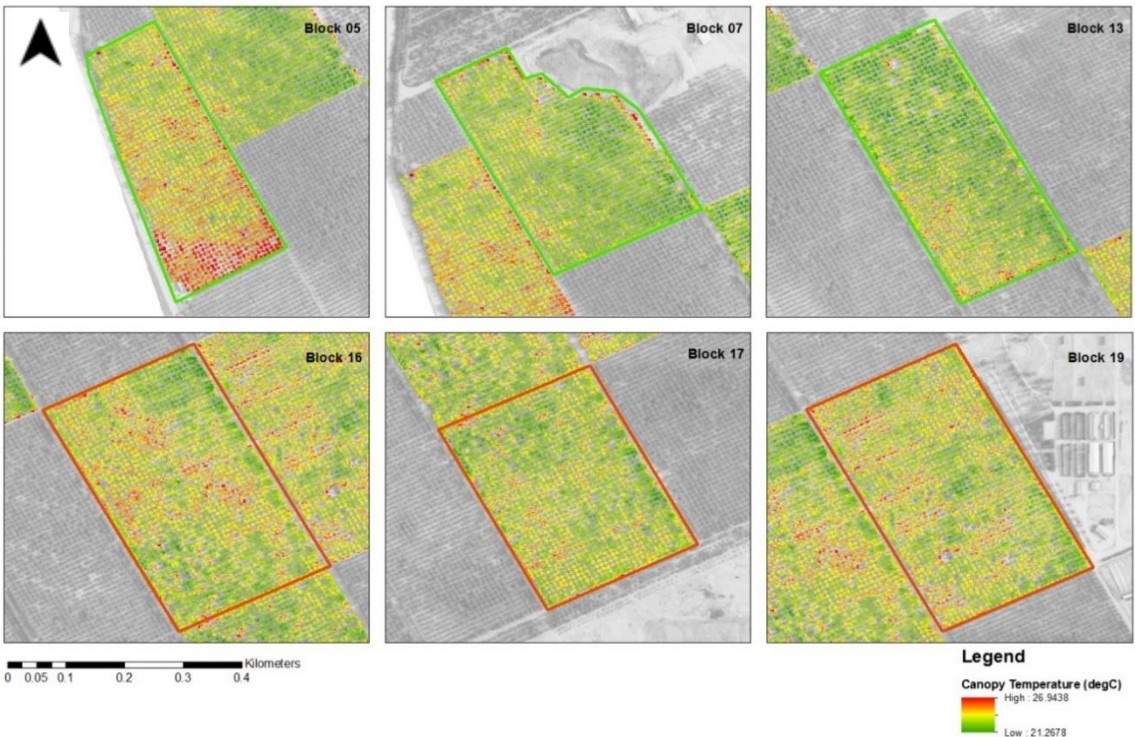

**Figure 7.** Temperature of the extracted canopy pixels representing individual trees for the test blocks.

Figure 8 shows the values for HAR in each block, where height is divided by canopy area. The healthy test blocks (5, 7 and 13) appear to have mostly values between 0.15 and 0.2 and Blocks 16 and 17 have more tree polygons with a larger height area ratio value, although one area of Block 5 contains trees with larger HAR values. This is to be expected as it indicates that there are more trees with abnormally small canopies, which one would expect in an unhealthier block. However, Block 19 has low HAR values, and in fact looks healthier than some of the "healthy" blocks. The trees in this block are shorter than in the other blocks, and the canopy area is similar, and thus it has the lowest mean height area ratio of all the blocks.

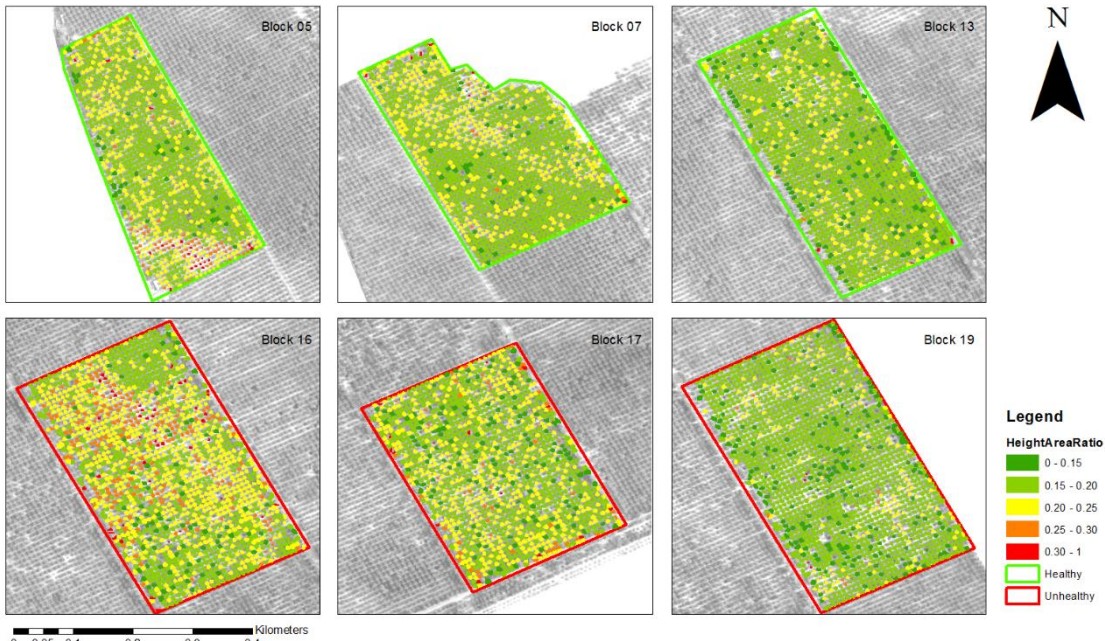

**Figure 8.** The height area ratios of the derived tree polygons representing individual trees for the test blocks.

Based on Figures 7 and 8, it is evident that within the blocks there is a lot of variation. This is especially true for Block 5, where there is a clear distinction in temperature and height between the north and south of the block.

ANOVA analysis was conducted on 10 randomly selected (within a height threshold; see Section 2.2.3) 'healthy' trees and 10 'unhealthy' trees per block, in order to evaluate whether the two factors of assumed health and the block containing the tree have an effect on tree temperature, and also whether there is interaction between the two (Table 4). There is a main effect of both health (probability ($p$) value = 0.000611) and block number ($p$ value = 1.06E-14), and there is no interaction effect ($p$ value = 0.834). The statistically significant outcome for block number indicates that the differences in the mean temperature of all the trees in a block are large enough to be considered as different. When comparing the mean temperature of all healthy trees and all unhealthy trees there is a statistically significant difference.

**Table 4.** Results of ANOVA analysis for temperature in relation to block number and the assumed block health.

|  | **Df** | **Sum Sq** | **Mean Sq** | **F Value** | **Pr (>F)** | **Significance** [1] |
|---|---|---|---|---|---|---|
| Block No. | 5 | 20.515 | 4.103 | 21.161 | 1.06E-14 | *** . |
| Health | 1 | 2.417 | 2.41 | 12.465 | 0.000611 | *** |
| Block No.: Health | 5 | 0.407 | 0.081 | 0.419 | 0.834311 |  |
| Residuals | 108 | 20.94 | 0.194 |  |  |  |

[1] Significance levels: 0 '***'; 0.001 '**'; 0.01 '*'; 0.05 '.'; 0.1 '·'; 1.

The results of post-hoc Tukey's HSD test reveal which mean differences contribute most to the statistical significance of the ANOVA results. A subset of results in Table 5 show six comparisons between healthy and unhealthy trees for each test block. When looking at the $p$ value, we see that they are all greater than a significance level $\alpha$ = 0.05, indicating no statistical significance. This suggests that when comparing the means of healthy and unhealthy trees within each block there are no statistically significant differences between these means. The differences in means of healthy and unhealthy trees range from 0.1 °C to 0.45 °C. This leads to the conclusion that the significant results seen in Table 4 are

a result of comparing trees from different blocks. However, in all cases, we do see that the unhealthy trees have a mean temperature higher than the healthy trees, which does match the hypothesis that unhealthy trees have higher temperatures due to stressors [6,17].

**Table 5.** Selection of Tukey's HSD comparing mean temperatures of selected healthy (*n* = 10) and unhealthy trees (*n* = 10) per block.

| | Mean Temperature (°C) | | | *p* Value |
|---|---|---|---|---|
| | Unhealthy | Healthy | Difference | |
| Block 5 | 24.37052 | 24.10113 | 0.26939 | 0.967108 |
| Block 7 | 23.30163 | 23.04954 | 0.25209 | 0.979962 |
| Block 13 | 23.02733 | 22.77498 | 0.25235 | 0.979803 |
| Block 16 | 23.80391 | 23.4071 | 0.39681 | 0.682276 |
| Block 17 | 23.40484 | 23.31995 | 0.08489 | 0.999999 |
| Block 19 | 23.68493 | 23.23745 | 0.44748 | 0.50232 |

### 3.3. Individual Tree-Level Analysis

Individual tree analysis has been carried out in order to compare the temperature and height values per tree to the results of the vegetation analysis. Block 11 has been selected for further analysis as there are a wider variety of trees, such as those with wider and smaller canopies. The 100 randomly selected test trees are shown in Figure 9, overlaid on the RGB image, the Chlorophyll Index Green (CIG-Table A1) vegetation index and thermal imagery. The hyperspectral image has geometrical distortion across the whole image, and it is especially notable in the northern region, and along the edges. However, when comparing with the visual (RGB) image, definite patterns are visible when comparing the vegetation index values and the tree canopy density that is visible in Figure 9a. In the south of the block, for example, there is more dense vegetation, and on the right side on the bottom there are patches of sparser vegetation. The regions of sparse vegetation in the top right and dense vegetation in the central region are also visible in the thermal imagery. The randomly selected test points cover a good range of these areas, with points in both the dense and sparse regions of canopy.

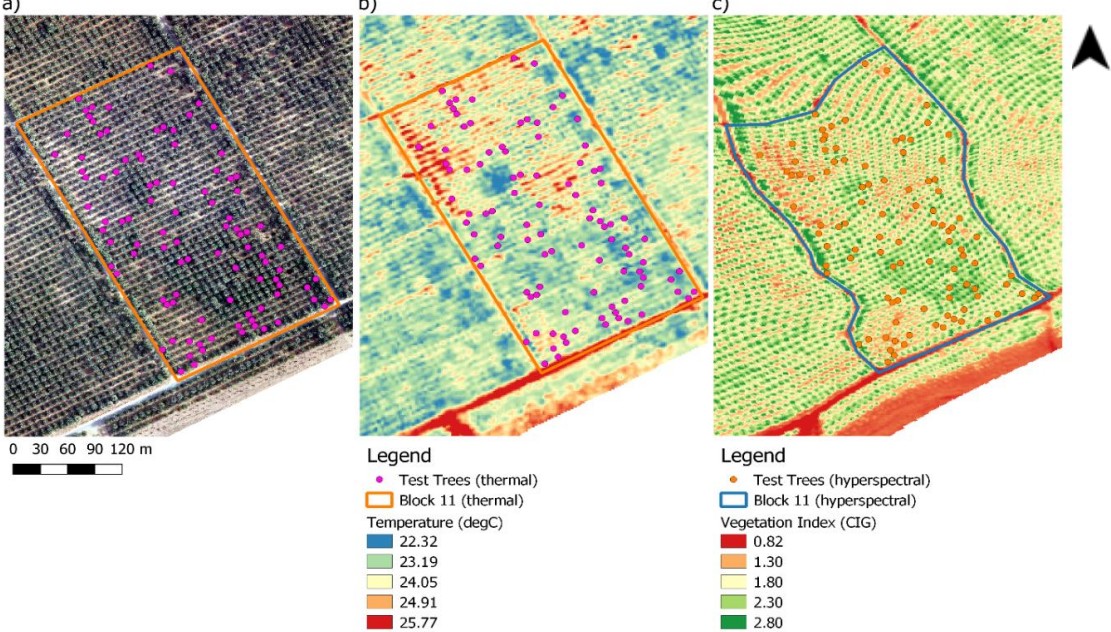

**Figure 9.** Illustrations of the selected test trees in Block 11 for (**a**) the visual RGB imagery, (**b**) the thermal imagery, and (**c**) the hyperspectral imagery represented as calculated Chlorophyll Index Green (CIG) (Table A1).

The correlation analysis between the vegetation indices and the height and temperature data show some interesting results. The R-squared values reach a maximum of 0.313 between the vegetation index red-edge position (REP) and temperature, and 0.253 between REP and height (Figure 10). The vegetation index Vogelmann red edge index (VOGI) also has a relatively strong correlation value with both temperature ($R^2 = 0.227$) and height ($R^2 = 0.213$) (Figure 10) and these indices both use bands in the red edge positions. Green NDVI (gNDVI) and Chlorophyll Index Green (CIG) also show relatively high correlation values (Table A2). These R-squared values are quite low in general, but as Figure 10 shows there is a trend between the two most highly correlated VIs (REP and VOGI) plotted against temperature and height for the 100 test trees. It is evident that temperature has a negative correlation with the VIs, indicating that at higher temperatures, these VI values tend to be lower. Height has the opposite correlation—taller trees have higher VI values.

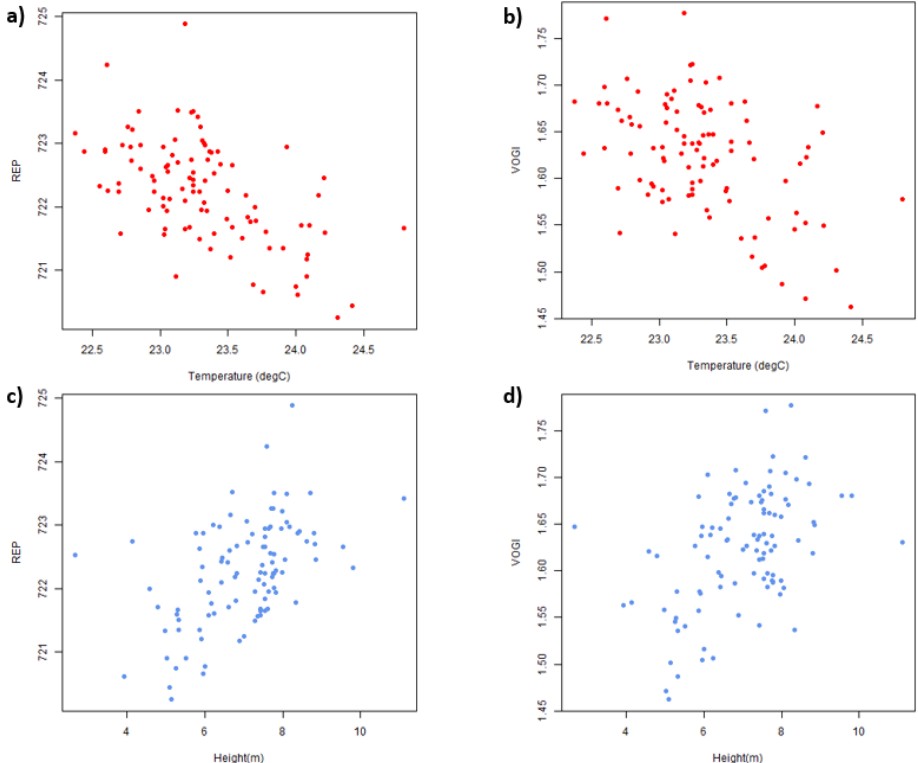

**Figure 10.** Correlation analysis for 100 test trees in Block 11: (**a**) and (**b**) show the correlation between temperature and two selected VIs, red-edge position (REP) and Vogelmann red edge index (VOGI), and (**c**) and (**d**) show the relationship between these indices and height.

In Figure 11 the temperature and height of the test trees are plotted overlaying the VOGI vegetation index image. It is evident that the regions with lower VI values tend to have higher temperatures and shorter trees; the opposite is generally true in regions with high VI values.

To get a clearer impression of the relation between the temperature and height values and VIs, the values have been visually displayed in relation to the VOGI vegetation index. It can be noted that the regions with lower VI values tend to have higher temperatures and shorter trees; the opposite is generally true in regions with high VI values.

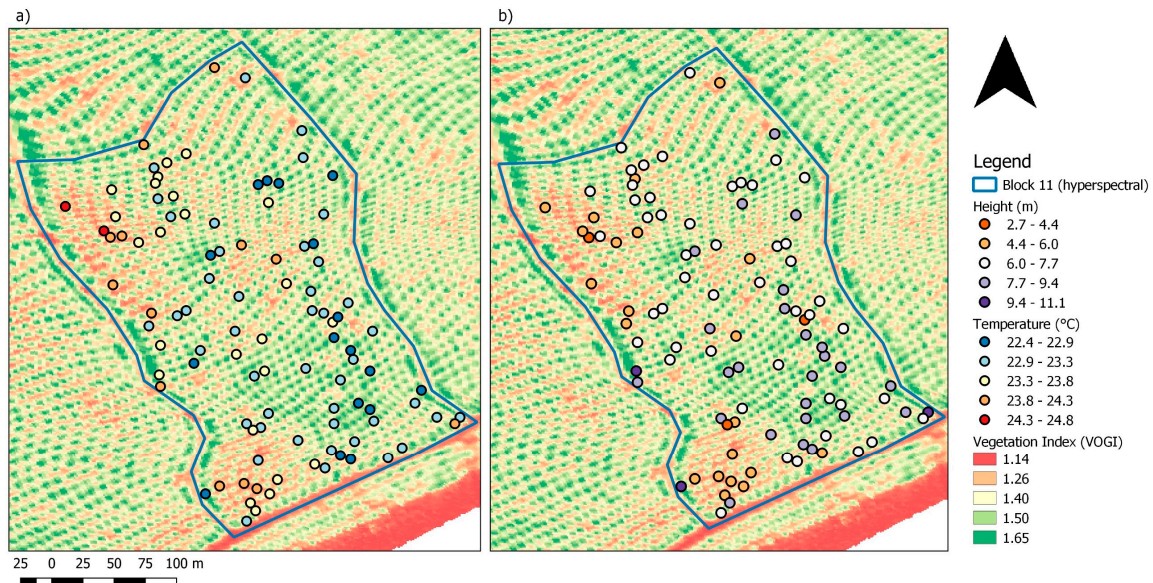

**Figure 11.** Illustration of the spatial patterns between tree-based point observations of (**a**) temperature and (**b**) height, with the vegetation index VOGI visualized as background map for comparison.

## 4. Discussion

This study explored various options for assessing vegetation health of date palm plantations using a combination of remote sensing data sources, specifically LiDAR, thermal, hyperspectral and visual RGB imagery. In order to differentiate between more general stressors and those that only affect individual palms, the plantations have been assessed on both a block and individual tree level. Although no direct field data was available for validation at tree level, exchange with the plantation manager provided support for the identified spatial patterns. The main contribution that this research can make is as a suggestion for methods to be used in further research. That being said, the preliminary results do show an ability to successfully differentiate between groups of trees and can be compared to earlier studies [6,16,17,19].

### 4.1. Exploring the Main Findings of the Study

On a block level, it is evident that the remote sensing images do highlight differences between the blocks; the results of the thermal box plots, for example, show up to a maximum of 1 °C difference between the blocks (Figure 5). These box plots show variation across the data—the spatial maps in Figures 7 and 8 illustrate how this variation is spatially distributed. Block 5 is a clear example, as it shows two very distinct areas in the north and the south of the block, where the trees are both much warmer and shorter. These inter-block variations can be seen to a lesser extent within each block. Therefore, it could be useful to add a further level to the current analysis, namely the development of management zones within the blocks that are based on tree characteristics.

ANOVA analysis was carried out to compare the temperature of individual healthy and unhealthy trees across each block, a method that has been applied in previous studies [6,17]. In the current study, the ANOVA results were statistically significant (Table 4) when comparing block number (*p* value = 1.06E-14) and tree health (*p* value = 0.000611) independently of one another. Tukey's HSD test (Table 5) shows the most significant differences come from comparing the trees in different blocks. There is no statistically significant result when comparing means of healthy trees in one block with means of unhealthy trees in that same block. The differences in means when comparing the unhealthy and healthy trees of one block range between 0.1 and 0.45 °C, which may not be practically significant to the extent that significant differences between trees can be identified based solely on temperature. When comparing the individual tree differences found in this study with those found

by Cohen et al. (2012), who noted that trees under an 80% water deficit were warmer than the control treatment by approximately 1 °C following four months of the assigned water treatment, the differences in temperature seen in the current study seem notable [6]. Cohen et al. (2012) used comparable thermal images with a resolution of 1.8 m [6]. Golomb et al. (2015), using thermal images with a resolution of 0.5 m, also found statistically significant results when comparing mean Crop Water Stress Indices (CWSI) values of control trees and infested trees using ANOVA, with a statistical significance of $p = 0.001$ [17]. When looking at the absolute temperature difference, a preliminary study by Soroker et al. (2013) noted a mean temperature difference of 1.5 °C between trees infested with *R. ferrugineus* and control trees [16].

Spatial resolution for the thermal data was 1.8 m, which is comparable to the study of Cohen et al. (2012), but lower than for Golomb et al. (2015), where the pixel size was 0.5 by 0.5 m [6,17]. This spatial resolution could have several implications for the results. Based on the RGB images, a large palm tree canopy is approximately 6–8 m in diameter and is covered by approximately 9 to 16 pixels for 1.8 m resolution. However, due to the leaf structure of the date palm canopy there are areas where the background soil will be visible in the pixel, and temperature of the background soil will also be included in the pixel value—it is probable that only one or two pixels include only pure canopy. In studies looking for a large-scale pattern such as the effect of irrigation on an entire plantation, these changes could be visible at a resolution of 1.8 m because every tree in a block is affected by the lowered irrigation [6]. However, for analysis on an individual tree level this resolution may be too coarse to detect whether a tree has a markedly different temperature in comparison to the surrounding trees. Therefore, a maximum resolution of 0.5 m such as in the thermal images acquired by Golomb et al. (2015) might be required to be able to effectively assess individual tree stressors [17].

Considering the results of the hyperspectral data assessment several conclusions can be drawn for individual tree analysis based on the vegetation index analysis. In comparison with other studies, VOGI was one of the most indicative VIs of palm stress in the study by Shafri et al. (2012) on oil palms [19]. gNDVI was also found to identify *R. ferrugineus* infestation of date palms [18]. These were also two of the most indicative VIs when looking at correlations on an individual tree level in this study (Figure 10), indicating that VIs which focus on red-edge and green bands may be most suitable for date palm health analysis.

### 4.2. Added Value of Combined Datasets and New Indicators

Evaluating the combination of remote sensing data sources recorded at the same timestep makes this study unique in the sphere of date palm management. Several studies have shown that the combination of remote sensing sources can provide multivariate indicators with improved characterisation of the vegetation health status [20,26].

From a methodological perspective, the ability to combine datasets has proven to be very relevant when identifying trees. Using the tree objects derived from the LiDAR data to select the canopy pixels for temperature analysis provided an alternative to the method proposed in Cohen et al. (2012), which utilised the watershed algorithm to extract canopy pixels [6]. The LiDAR data again proved useful for temperature data analysis as the pixels below a certain height could be removed to reduce the chance of soil pixels being included in the analysis. A drawback is that an extra sensor is required, which will add costs to the data acquisition process. The high-resolution RGB data was compared with the outcome of the canopy extraction method to check whether the trees were delineated well (Figure 3).

Correlation analysis between thermal and hyperspectral datasets showed promising relationships (Figure 10). The expected trends are visible, with negative correlation between temperature and several of the VIs. A positive correlation was observed for height and the VIs. Whilst the R-squared values between the variables are quite low, there is a clear spatial pattern between the indicators and VIs as seen in Figure 10. This suggests that these data, especially thermal and hyperspectral imagery, could be combined to identify stressed trees. The temperatures of the trees range from 22.5 °C to 24.5 °C (Figure 10), a difference of 2 °C. Cohen et al. (2012) were able to illustrate differences in temperature

of stressed and healthy palms of 1 °C [6]. This provides an indication that using thermal data could prove useful for diagnosis of individual tree health. Based on the correlation results, the hyperspectral data should be useful for reinforcing this diagnosis.

Based on the results of the tree point extraction using NDVI, it can be concluded that this method has high potential for identifying tree objects. Over 90% of trees were identified based on a comparison of field measured trees and NDVI, assuming accurate tree detection (Section 3.1). Misidentifications occur along the edges of the block where the trees are often close together and individual tree canopies are not distinct. As these results are based on distorted data one can assume that, when using a geometrically corrected dataset, identification accuracy will be higher. Using this identification method could prove to be an alternative to expensive LiDAR techniques. It should be noted that one of the reasons this method works well is because the canopies are distinct in a data palm plantation; however, in denser canopies such as the edges of date palm blocks, identification may be less accurate.

In this study, the height area ratio (HAR) index was used in order to assess the health of the blocks. When conducting analysis on the height of the trees, it became apparent that there were large differences in height of trees in certain blocks. For example, Block 19 had lower trees than the other blocks (Figure 6). From this information, one could conclude that either Block 19 is less healthy than the other blocks, or that the trees are younger in that block. Therefore, adding canopy area as a variable when considering tree health adds another consideration to the analysis. By determining the ratio between height and canopy area, one can determine whether a tree has an abnormally small canopy in relation to height—this could be an indication that the tree is experiencing growth limitations. In a study by Shendryk et al. (2016), a width height ratio was included amongst hyperspectral and LiDAR indicators to determine the most "useful" indicator for assessing tree health using Principal Component Analysis (PCA) [20]. Of the indicators derived from the recorded tree height, the width height ratio was the most important in their study. The conclusion that taller trees have smaller canopies in comparison to shorter trees indicates that using HAR alone does not give an accurate representation of tree health, as the growth of the tree affects this value for reasons other than health. Therefore, using an index that normalises height area ratio based on the height of a tree could prove more meaningful when making conclusions about tree health.

*4.3. Data Limitations*

As has been discussed in the Section 2.2.1 on preprocessing, there are some issues with data quality, especially for the hyperspectral dataset used in this study. Both the geometric distortions and the unknown scaling factor mean that the values used in the analysis might not be fully representative and certainly cannot be directly compared to other hyperspectral datasets that could be used in the future. For example, one step involved using bilinear interpolation for geo-correction, which resulted in multiple raster points with the same value because the hyperspectral image was effectively stretched to fit the corrected dataset.

Due to the distortions of the hyperspectral analysis, it was not possible to compare the trees on an individual level without manually matching the points. This meant that a relatively small sample size of 100 trees has been used as representative of over 10,000 trees. It is also possible that errors have been made when manually identifying the corresponding trees, as in some cases it was difficult to count rows and columns accurately within the distorted data.

Additionally, there are issues within the thermal dataset; specifically, for some limited areas blurring was noticed. This is due to the ortho-mosaicking technique used to merge the snapshot images. Trees are not identifiable in some regions, meaning that extracting the temperature per tree will result in values that are not an accurate representation of tree temperature. This was considered when selecting a block for individual tree analysis carried out in Section 2.2.4, as it was important that the blurring effect was minimal.

Finally, without ground truth data, it is not possible to determine a measure of actual tree health. This means that all assumptions made about tree health are based on indirect field observations and

the relative health indicated by the RGB imagery (Table 2). The remote sensing datasets have been compared in various ways to determine whether the expected relations for healthy and unhealthy trees can be seen, and in some cases the initial assumptions were correct. However, as was seen in Figure 5, Block 5 had a much higher temperature than expected despite being considered as a healthy block. This suggests that there were factors not considered in classifying the blocks, and further field information was required. Following the analysis, further field information was provided, and the farm manager indicated that Block 5 had relatively poor soil quality in the southern region. Validation with field data is an essential step for future studies to explore the methods discussed in this paper further. As a result of the limited field data, especially on an individual tree level, the conclusions that can be made are limited to identifying that there are differences in date palms or date palm groups, and these differences cannot be attributed to a specific stressor.

### 4.4. Future Developments

Overall, this study reiterates the findings of earlier research indicating that different remote sensing datasets could prove useful for management of date palm plantations [6,16,17,19]. Thermal and hyperspectral data are the most promising, especially since identifying trees using NDVI is comparable to the results of LiDAR analysis. For future studies the opportunities of high-resolution RGB imagery to identify variation in palm canopy geometry could be investigated by using a combination of RGB based vegetation indices [27] and object-based image analysis.

Remote sensing data will prove most useful when provided as part of an integrated plantation management platform, where field data is being collected as the flight is taken, and ideally field data will be available over an extended period of time. The imagery could provide useful insights at all steps of an integrated system, including; field registration, field surveys, data acquisition, and monitoring. For example, the step of tree identification using NDVI or LiDAR would be useful for field registration. A topic that could be interesting to explore would be identifying within-block parcels with differing characteristics to be classified as management zones, which would allow for the adoption of precision agriculture techniques. Regular monitoring using a combination of satellite and aerial, or even drone imagery [28], potentially using the methods as explored in this article, could add value to the current monitoring systems that are in place. Based on a time-series analysis approach, changes in vegetation reflectance properties can be indicative of palm health, production and disease infestation processes over the growing season. For example, Jimenez-Brenes et al. [29] showed how multi-temporal UAV-based images can be adopted to support pruning management in olive tree orchards. The ground monitoring system should also be well designed and could include ground sensors [30] and standardised methods of recording tree characteristics. Based on these spatial-temporal data, spatial analytics and geostatistical functions can be adopted to evaluate infestation processes of pests and diseases both at farm and regional scale-level [31]. A design based on the DateGIS platform approach could prove useful, as it aims to integrate imagery data with a ground data recording system where farm employees can update information about specific trees in the field [32].

### 5. Conclusions

This research explored the options of using multiple types of high-resolution remote sensing imagery in order to assess its potential for detecting various health aspects of date palm plantations. The factors which affect health were considered on two levels; those that impact large areas of the plantation such as a block, and those that affect the individual trees. Based on the spatial variation within blocks, it is suggested to also consider a third level in future studies; namely, areas within blocks with similar characteristics. Despite issues with the hyperspectral and thermal data preprocessing and quality, the methods developed in this study provide new options for future analysis and indicate that remote sensing data could aid plantation management and supplement precision agriculture techniques. Specifically, using a combination of high resolution thermal and hyperspectral imagery

can give an indication of individual tree health, and by using these indicators together it could be possible to define a status for each tree. There are many new developments to be made in this area of research—combining remote sensing with detailed field data could provide an early indication of *R. ferrugineus* infestation; and comparing flights at different time steps could provide insight into the rates of change of date palm health. Overall, this study adds to the currently small body of research regarding using remote sensing imagery for date palm plantation management by using a combination of data sources and by providing suggestions for future research in this topic.

**Author Contributions:** Conceptualization: M.M., L.K., L.B.; Methodology: M.M. and L.K.; Software: M.M.; Formal analysis: M.M..; Writing—Original draft preparation: M.M..; Writing—Review and editing: L.K and L.B.; Visualisation: M.M.; Supervision: L.K. and L.B.; Project administration: L.B.; Funding acquisition: L.B.

**Funding:** This research was funded by the European Space Agency (ESA) under contract 4000116766/16/NL/US.

**Acknowledgments:** The authors want to acknowledge Mohamed Ali Bob, farm manager of the Al Mohamadia date farm (www.mohamadia.com.sa/en/) for his valuable support on date farm management and ground truthing feedback, and TAQNIA ETSC for provisioning and operating of the Diamond DA42 MPP GEOSTAR aircraft. Further, the authors want to acknowledge Harm Bartholomeus (Wageningen University and Research) for his support on LiDAR data preprocessing, Juha Suomalainen (Finnish Geospatial Research Institute) for support with the preprocessing to generate a mosaic for the thermal dataset, and Daniel Iordache (VITO) for support in preprocessing of the hyperspectral dataset.

**Conflicts of Interest:** The authors declare no conflicts of interest.

## Appendix A

**Table A1.** Formulae for all hyperspectral vegetation indices used for the hyperspectral analysis. $R_n$ refers to the band number used.

| Vegetation Index | Formula | Reference |
|---|---|---|
| SRI | $\frac{R_{800}}{R_{670}}$ | [19,33] |
| NDVI 800 | $\frac{R_{800} - R_{660}}{R_{800} + R_{660}}$ | [19,34] |
| MSR | $\frac{R_{800}/R_{670} - 1}{(R_{800}/R_{670})^{-1} + 1}$ | [19,35] |
| NDVI 750 | $\frac{R_{750} - R_{660}}{R_{750} + R_{660}}$ | [36] |
| VOGI | $\frac{R_{740}}{R_{720}}$ | [19,37] |
| CIR | $\frac{R_{780}}{R_{710}} - 1$ | [38,39] |
| gNDVI | $\frac{R_{801} - R_{550}}{R_{801} + R_{550}}$ | [18,40,41] |
| CIG | $\frac{R_{780}}{R_{550}} - 1$ | [38] |
| PRI | $\frac{R_{531} - R_{570}}{R_{531} + R_{570}}$ | [42,43] |
| Optimum NDVI | $\frac{R_{738} - R_{610}}{R_{738} + R_{610}}$ | [19] |
| SIPI | $\frac{R_{800} - R_{445}}{R_{800} + R_{680}}$ | [18,44] |
| MCARI/OSAVI [705,750] | $\frac{[R_{750} - R_{705} - 0.2(R_{750} - R_{550})](R_{750}/R_{705})}{(1+0.16)(R_{750} - R_{705})/(R_{750} + R_{705} + 0.16)}$ | [45–47] |
| TCARI/OSAVI [705,750] | $\frac{3[R_{750} - R_{705} - 0.2(R_{750} - R_{550})(R_{750}/R_{705})]}{(1+0.16)(R_{750} - R_{705})/(R_{750} + R_{705} + 0.16)}$ | [46,47] |
| REP | $700 + 40\frac{R_{670} + R_{780}/2 - R_{700}}{R_{740} - R_{700}}$ | [48] |

## Appendix B

**Table A2.** Results of R-squared analysis of the vegetation indices contained in Table A1 and their relations to temperature and height.

| Vegetation Index | Temperature (R-Squared) | Height (R-Squared) |
|:---:|:---:|:---:|
| REP | 0.313 | 0.253 |
| VOGI | 0.227 | 0.213 |
| gNDVI | 0.206 | 0.189 |
| CIG | 0.196 | 0.204 |
| CIR | 0.196 | 0.202 |
| TCOS750 | 0.191 | 0.103 |
| SIPI | 0.137 | 0.161 |
| SRI | 0.119 | 0.147 |
| NDVI800 | 9.106 | 0.146 |
| NDVI750 | 0.088 | 0.136 |
| MSR | 0.08 | 0.14 |
| NDVI730 | 0.071 | 0.112 |
| MCOS750 | 0.054 | 0.134 |
| PRI | 0.001 | 0.014 |

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
