# Peer review of "High-Resolution Multisensor Remote Sensing to Support Date Palm Farm Management"

_agriculture, doi:10.3390/agriculture9020026_

Reviewer 1 Report

This study used high-resolution multi-sensor remote sensing data to support date palm farm management. In general, motivation is well-emphasized and analysis of experimental results is convincing. However, there are some issues that should be considered, especially related to remote sensing data processing.

The most critical problem in this study is to convert the degrees Celsius, which is just divided by a factor of 0.19 to match the grayscale value 255 to degrees Celsius 50. The converted value is not a real degrees Celsius. It would be some ways (equations or software) to get real degrees Celsius. Please check (webpage of VarioCAM or Internet search) and use the real degrees Celsius for experiments. All the experimental analysis related to the temperature should be reconstructed.

All the abbreviations should be defined their full names when they first mentioned in the manuscript (e.g., NDVI, ITC, LiDAR, etc.).

In this study, all the remote sensing data were used independently. It would be much more meaningful if those data are used together to estimate agriculture-related products.

It should be mentioned the number of spectral bands of hyperspectral dataset in the Table 1.

There are too severe distortions in Hyperspectral data to be used for experiments. There are some VIs derived by only RGB bands. Therefore, it would be a good alternative to use those RGB-based VIs instead of hyperspectral-based VIs that show severe distortions.

Figure 2: It would be better to show some enlargement figures of healthy and unhealthy blocks with the existence of red palm weevil.

There are some grammar errors and typos. Please carefully double check the entire manuscript.

Author Response

Please find our response to the review comments in the attached Word Document.

Reviewer 2 Report

In Figure 1,3,4,7,9,11 the sign of the “North” should be added.

Author Response

(The authors gave the same response as above.)

Reviewer 3 Report

This paper show the ability of remote sensing data to be a supplement technic to help plantation management. As an example, authors show that, using a combination of high resolution thermal and hyperspectral imagery one can give an indication of individual tree health.

The results presented are promising and the authors propose further research to develop methods using combination of remote sensing data.

Furthermore authors address two aspects in the conclusion that they could have detailed them more in the paper. The first idea deals with the useful combination of remote sensing with in-situ data measurement that could provide, for example, an early indication of red palm weevil infestation. The second idea is based on the comparison of data extracted from different flights (airborne remote sensing data) at different time steps. This latter could provide useful information about changes of date palm health in the plantation.

Some remarks :

Figure 1 : improve text caption

Table 1 : skip lines to separate the different fields and make the table easier to read

Line134

“The image was received as a (8 bit) greyscale image with 256 levels, and to convert the values into a relative approximation of degrees Celsius, the image was multiplied by a factor of 0.1953125, which is the expected maximum temperature in the region (50°C) divided by the value 256.”

256 is the numerical range so the factor value computed is right if the thermal range is 50°C. Does the minimum temperature value is 0°C ?

Table 2 : How do you compute the percent distorted

Line 262 Why did not you do an array equivalent to Table 3 (LiDAR data) for NDVI image results ?

Line 339 Please remind to reader that p value = probability value

Table 4 why did you use significance codes and not values?

Figure 11 Can you avoid putting the explanations in the title of the figure. 

The color indications in the circles for each tree are difficult to visualize

Line 444 “spatial resolution could have several implications for the results. Based on the RGB images, a large palm tree canopy is approximately 6-8m in diameter and is covered by approximately 9 pixels for 1.8m resolution.”

For 1.8m resolution, the pixel surface is 3,24 m^2. A 8 meter diameter canopy has a surface of 50,24 m^2 so instead of 9 pixels you can write “… approximately 10 to 16 pixels…” 

Author Response

Please find our response to the review comments in the attached Word Document.

Round  2

Reviewer 1 Report

I still have some concerns that you didn't use the real temperatures for the analysis. Except of it, most of the issues that I had have been addressed in the revised manuscript.